# miRNA normalization enables joint analysis of several datasets to increase sensitivity and to reveal novel miRNAs differentially expressed in breast cancer

**Shay Ben-Elazar**[1,2]*, **Miriam Ragle Aure**[3,4]*, **Kristin Jonsdottir**[5,6], **Suvi-Katri Leivonen**[7], **Vessela N. Kristensen**[3,4,8,9], **Emiel A. M. Janssen**[5,6], **Kristine Kleivi Sahlberg**[3,10], **Ole Christian Lingjærde**[3,11], **Zohar Yakhini**[2,12]*

1 School of Computer Science, Tel-Aviv University, Tel-Aviv, Israel, 2 Department of Computer Science, Interdisciplinary Center, Herzliya, Israel, 3 Department of Cancer Genetics, Institute for Cancer Research, Oslo University Hospital, Oslo, Norway, 4 Department of Medical Genetics, Institute of Clinical Medicine, University of Oslo and Oslo University Hospital, Oslo, Norway, 5 Department of Pathology, Stavanger University Hospital, Stavanger, Norway, 6 Department of Chemistry, Bioscience and Environmental Engineering, University of Stavanger, Stavanger, Norway, 7 Helsinki University Hospital Comprehensive Cancer Centre and University of Helsinki, Helsinki, Finland, 8 Institute for Clinical Medicine, University of Oslo, Oslo, Norway, 9 Department of Clinical Molecular Biology and Laboratory Science (EpiGen), Division of Medicine, Akershus University Hospital, Lørenskog, Norway, 10 Department of Research, Vestre Viken Hospital Trust, Drammen, Norway, 11 Centre for Cancer Biomedicine, University of Oslo, Oslo, Norway, 12 Department of Computer Science, Technion–Israel Institute of Technology, Haifa, Israel

* shay.benel@gmail.com (SBE); Miriam.Ragle.Aure@rr-research.no (MRA); zohar.yakhini@gmail.com (ZY)

**Data Availability Statement:** All relevant data are within the manuscript and its Supporting Information files.

## Abstract

Different miRNA profiling protocols and technologies introduce differences in the resulting quantitative expression profiles. These include differences in the presence (and measurability) of certain miRNAs. We present and examine a method based on quantile normalization, Adjusted Quantile Normalization (AQuN), to combine miRNA expression data from multiple studies in breast cancer into a single joint dataset for integrative analysis. By pooling multiple datasets, we obtain increased statistical power, surfacing patterns that do not emerge as statistically significant when separately analyzing these datasets. To merge several datasets, as we do here, one needs to overcome both technical and batch differences between these datasets. We compare several approaches for merging and jointly analyzing miRNA datasets. We investigate the statistical confidence for known results and highlight potential new findings that resulted from the joint analysis using AQuN. In particular, we detect several miRNAs to be differentially expressed in estrogen receptor (ER) positive versus ER negative samples. In addition, we identify new potential biomarkers and therapeutic targets for both clinical groups. As a specific example, using the AQuN-derived dataset we detect hsa-miR-193b-5p to have a statistically significant over-expression in the ER positive group, a phenomenon that was not previously reported. Furthermore, as demonstrated by functional assays in breast cancer cell lines, overexpression of hsa-miR-193b-5p in breast cancer cell lines resulted in decreased cell viability in addition to inducing apoptosis. Together, these observations suggest a novel functional role for this miRNA in breast cancer.

**Funding:** Helse Vest (https://helse-vest.no/en) grant 911450 was received by KJ. This project has received funding from the European Union's Horizon 2020 Research and Innovation Programme (https://ec.europa.eu/programmes/horizon2020/en) under Grant agreement No. 847912. The funders had no role in study design, data collection and analysis, decision to publish, or preparation of the manuscript.

**Competing interests:** The authors have declared that no competing interests exist.

Packages implementing AQuN are provided for Python and Matlab: https://github.com/YakhiniGroup/PyAQN.

## Author summary

This work demonstrates a practical approach to the joint-analysis of multiple miRNA expression profiling datasets acquired with different measurement technologies. The use of different platforms in miRNA profiling can lead to major differences in results. In particular, some miRNA species are less amenable to detection and quantification by certain platforms or designs. Our approach, termed AQuN, combines quantile normalization with special attention to missing entities, to normalize miRNA expression across datasets, technologies, designs and platforms. As we show, our proposed approach uncovers patterns of interest that would not have emerged as statistically significant when analyzing the datasets individually or with other standard-practice normalization methods.

Amongst our findings, we noted a previously undocumented miRNA that is significantly over-expressed in samples from estrogen-receptor positive breast cancer patients as compared to samples from estrogen-receptor negative patients. We further investigated this miRNA, hsa-miR-193b-5p, and experimentally show, in cell lines, that its expression level impacts the viability of tumor cells. AQuN is available to the community in the form of Python and Matlab packages. The joint-processed data is also made available for further investigation.

## Introduction

microRNAs (miRNAs) are endogenous, small non-coding RNAs (~22 nucleotides) that bind to target-specific sites most often found in the 3'-untranslated regions (UTRs) of target messenger RNAs (mRNAs). Through this binding, miRNAs regulate gene expression by conferring inhibition of mRNA translation or mRNA degradation [1]. miRNA expression profiling is an important tool for studying tumor biology and classification and serves as a basis for potential diagnostic and prognostic assessments [2–4]. Increasing technological and economic viability of expression sampling methods has enabled the systematic study of miRNA expression in cohorts of hundreds of patients [5–7] and in several cancer types [8, 9]. On the other hand, inherent measurement noise coupled with complex causes of biological variability affect the statistical confidence in ascertaining consistent differences of low magnitude between populations when limited to small sample sizes. Absolute expression differences are not necessarily linearly correlated with downstream effects of the expressed miRNA, therefore subtle but consistent differences may be of greater biological importance.

Abnormal miRNA expression in breast cancer has been repeatedly associated with cancer proteins [10], molecular subtypes [11], progression [12–14] and prognosis [5]. For example, in one of the first genome-wide characterization studies of miRNA expression in breast cancer we identified 63 miRNAs differentially expressed between the two main clinically diverse groups of breast cancer, estrogen receptor (ER) positive and the ER negative tumors [11].

Combining experimentally measured data from multiple sources is both a challenging and a worthwhile endeavor. Statistical estimation theory formulates a relation between sample size and variance of estimate via the Fisher information that follows the chain rule for independent samples. The ability of statistical hypothesis tests to detect subtle, yet consistent and possibly

genuine, differences between populations is directly related to sample size and is quantified as a test's power [15, 16]. Increasingly larger power and statistical significance is hindered by sampling costs that can prohibit large sample sizes. This, in turn, leads to the incremental funding of repeated studies aiming to measure the same phenomenon. Follow-up studies tend to vary from their former, sometimes using newer or alternative experimental protocols, reagents and technologies, introducing batch differences between samples. Such a 'batching' design, inadvertently, introduces distinctions (batch effects) between samples that correlate with the batch and may overshadow subpopulation differences in their magnitude. Blindly testing for hypotheses on batch-collected dataset without taking such effects into account can lead to spurious and erroneous conclusions and can hide significant effects behind batch differences. In this work we address joint analysis of data batched using different miRNA profiling technologies that have been shown to have systematic differences [17, 18].

There are various approaches commonly used in practice to address the analysis of combined data containing batch effects. The authors of earlier works [19, 20] showed that applying standard, parametric, batch correction approaches may introduce bias from uneven sample sizes of the different groups and data idiosyncrasies. A recent study [21] applied a non-parametric approach for correcting case-control microbiome studies and showed that it compares favorably with former methods. Their method resembles ours, as we further illustrate below.

In this work we apply a non-parametric, quantile-based, batch normalization approach, Adjusted Quantile Normalization (AQuN). We use this method for jointly analyzing miRNA expression data in four breast cancer cohorts to obtain increased statistical confidence and power. We demonstrate that, coupled with appropriate non-parametric statistics, our normalization approach lowers the confounding impact of batch effects. We observe stronger statistical evidence of differential expression between ER positive and ER negative clinical groups in multiple miRNAs when compared to individually analyzing the cohorts. Moreover, our approach provides interpretable results and is advantageous to direct interpretation of the data, conducive to individual examination of findings, as demonstrated herein. Our differential expression analysis supports the use of AQuN by surfacing known cancer-related miRNAs, as well as providing evidence of potential new ones.

In particular, previous studies have showed hsa-miR-193b-3p regulates breast cancer migration [22] and can function as a metastatic suppressor [23]. Here we discovered that hsa-miR-193b-5p is significantly over-expressed in ER positive, compared to ER negative clinical groups. We propose that this difference is of functional significance and further show it leads to decreased cell viability and increased apoptosis.

## Results

We apply the Adjusted Quantile Normalization (AQuN) process to the datasets described in [5, 11, 24–26] and illustrate the benefit and effects of this normalization step as related to data properties and to various downstream analysis steps in the subsections below.

AQuN is a novel variant of quantile normalization which utilizes quantization in its normalization process, thereby offering an added degree of control over noise that affects sample ranking and is evidently prevalent in miRNA datasets. Details of this method are available in the Materials and Methods section. Below we illustrate AQuN's advantage over standard normalization methods in uncovering, otherwise nascent, signals in the joint dataset.

### Differential expression reveals novel breast-cancer associated miRNA

We performed a differential expression analysis comparing clinically relevant subgroups of breast cancer. We measured differential expression of a specific miRNA on a pair of sample

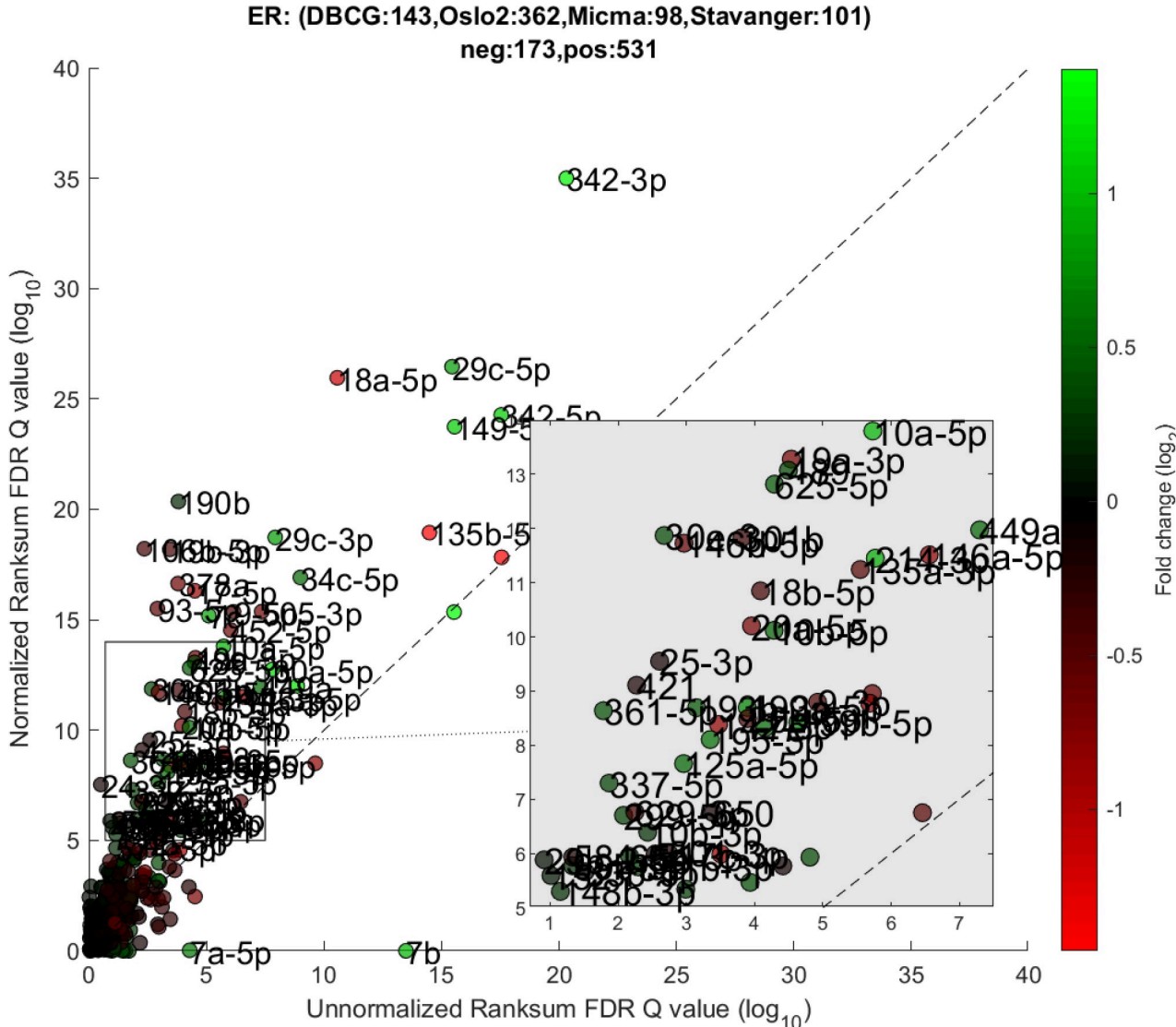

**Fig 1. Differential miRNA expression between ER positive and negative.** A scatter plot of differential expression p-values (-$log_{10}$, Wilcoxon Ranksum) for the unnormalized (x) vs normalized (y) joint dataset. Title contains sample size details and dataset distribution.

subpopulations (e.g. ER positive vs ER negative). Fold-change was defined as the ratio ($log_2$) between median expression of both sets. We applied Wilcoxon Rank-Sum (WRS) 1-tailed tests and resulting p-values were corrected across miRNAs using false discovery rates (FDR). Figs 1–3 showcases our differential expression analysis results for ER status. In Fig 1 scatter plot, we observe that the AQuN normalized dataset yields more significant results (lower Q-values) for most miRNAs (482/655). Fig 2 volcano plot illustrates that the increase in significance is not necessarily correlated with effect size (i.e. fold change), and that we gain confidence on lower effect sizes as anticipated by our power analysis (more details in the 4[th] paragraph of the Discussion section). In Fig 3 cumulative distribution function (CDF) plot we depict the overall trend of increased statistical significance, contrasted by even lower statistical significance that would be obtained from performing the differential expression analysis on each dataset separately (shown as dashed lines). In addition, we present the CDF plots that would be obtained

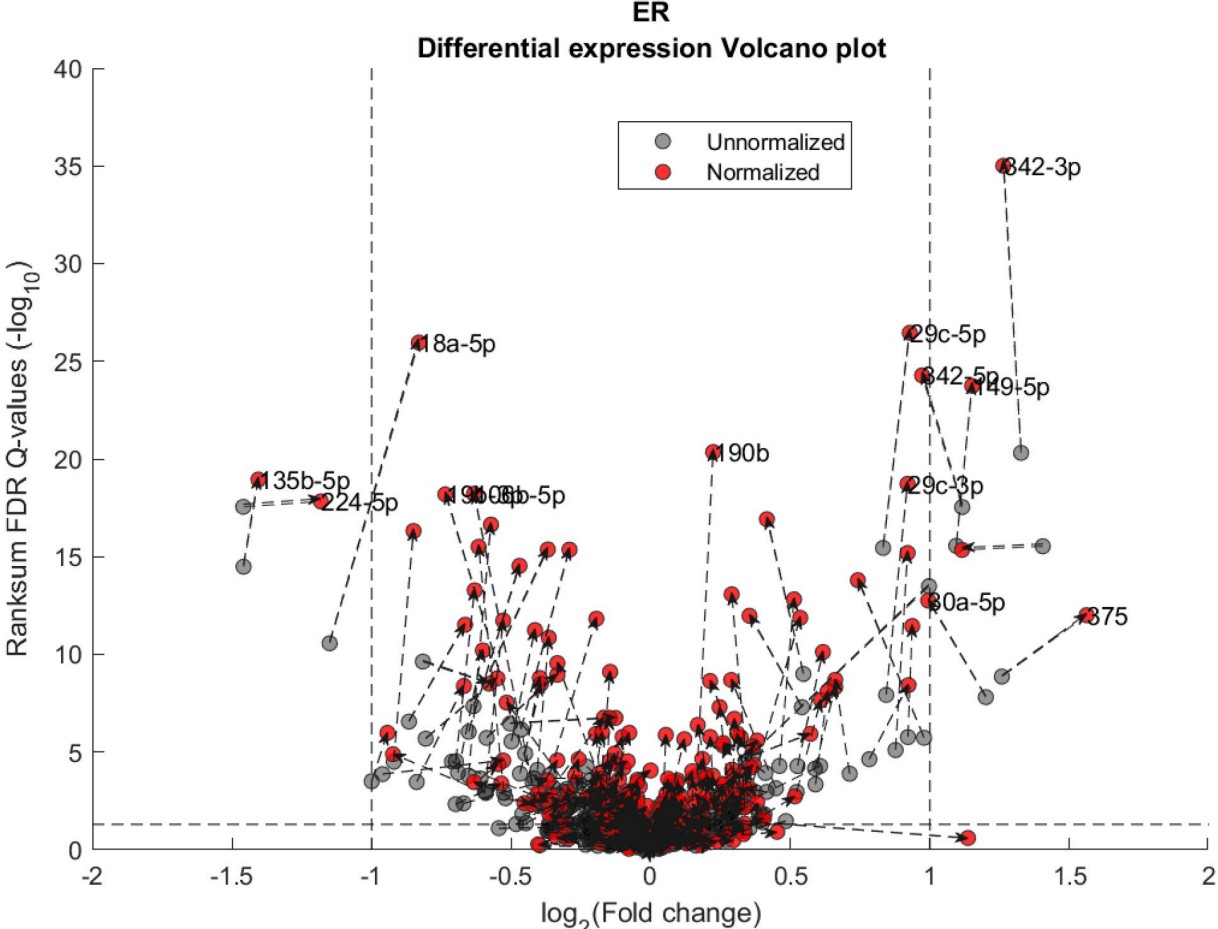

**Fig 2. Differential miRNA expression between ER positive and negative.** Volcano plot showing the fold change and corresponding Wilcoxon Rank-sum FDR corrected Q value ratio between the normalized and unnormalized datasets. Dashed arrow connects the unnormalized (gray circles) and normalized (red circles) results on a particular miRNA. High absolute values in X axis correspond to substantial difference in median expression between ER negative over ER positive samples (for a particular miRNA). High values in Y axis correspond to miRNAs that present substantial difference *after* normalization but not before. Low values in Y axis correspond to miRNAs that present substantial difference *before* normalization but not after. Vertical dashed lines represent a Fold change threshold of 2x ($\log_2(2) = 1$) and horizontal dashed lines represent a Q-value threshold of 0.05 ($-\log_{10}(0.05) \cong 1.3$).

by (individually) applying four commonly used normalization methods (shown as dotted lines). Evaluated normalization methods include:

- Mean ratio: scales each sample by $M(i,j) = \frac{M(i,j)}{Avg\ M(:,j)}$.

- Median subtraction: subtracts the median of each sample, then sets the minimum of each sample to the (global) minimum across samples. I.e.:

$M(i,j) = M(i,j) - Medial\ M(:,j).$ min $M(:,j) \leftarrow$ min $M(:,:)$

- Vanilla quantile: MATLAB's implementation of Quantile Normalization also known as Quantile Standardization [27].

- ComBat [28]: empirical Bayes batch effect mitigation employing a design matrix that includes dataset batching along with clinical labels and status of Tumor grade, Subtype, ER, PR, HER2 and TP53. We apply the QR decomposition [29] to mitigate any co-linearity in the design matrix.

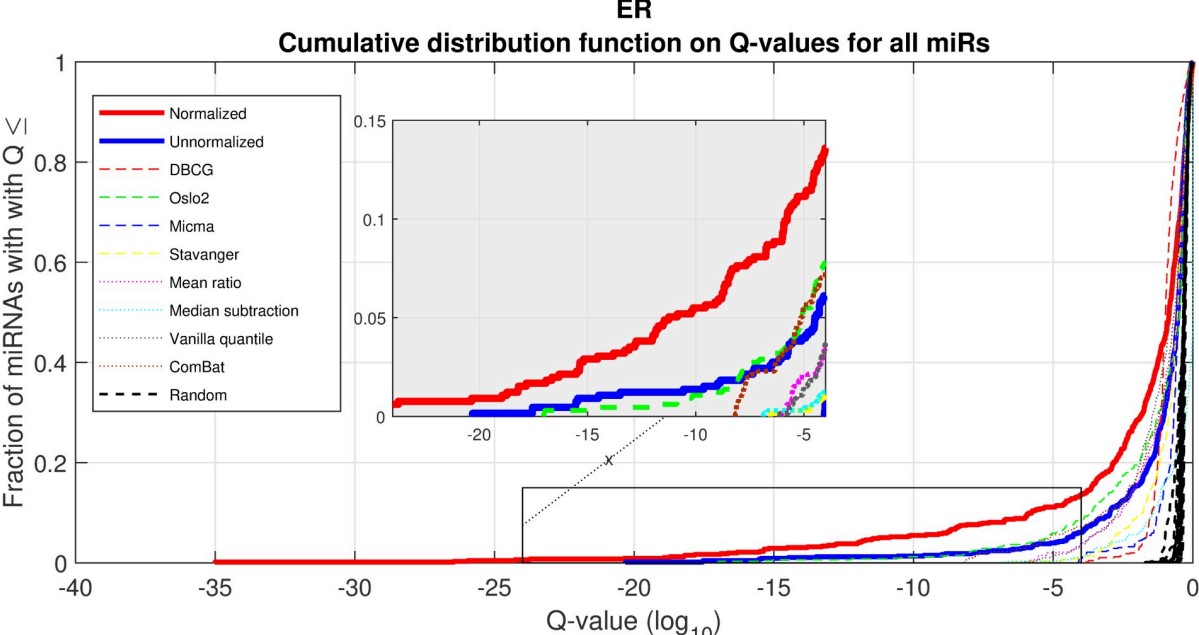

**Fig 3. Differential miRNA expression between ER positive and negative.** A CDF plot showing many more substantially differentially expressed miRNAs after normalization (red line) than before normalization (blue line), and substantially more than would be expected at random (compared to 20 random permutations of labels, dashed black lines). Also shown are dashed colored lines corresponding to each appropriate single-dataset Q values exemplifying the advantage of a joint-dataset analysis. Note that at $Q = 10^{-18}$ we can find 10 miRNAs under AQuN but none under other normalization approaches or per dataset analyses.

In Figs 4 and 5 we demonstrate the impact of normalization on single miRNAs (hsa-miR-190b and hsa-miR-18a-5p, accordingly) across samples and on their differential expression in the context of ER status. This is done by detailing expression values for each sample in the joint dataset prior to (top row) and following (bottom row) AQuN normalization. We present the medians of each clinical group (dashed horizontal lines) and a breakdown of how samples of both clinical groups are distributed when sorted by value and when compared to a uniform null model. This provides a qualitative view of the effect normalization has on both individual samples and datasets in the context of the investigated differential expression. Previous studies [30] have shown hsa-miR-190b to be linked to ER status and further suggested its use as a potential biomarker. Similarly, hsa-miR-18a-5p is an oncogene and prognostic biomarker [31]. As we have shown in the volcano plot in Fig 2, hsa-miR-190b would not have been identified as differentially expressed in ER positive vs negative samples prior to normalization. Similar plots for the top 40 differentially expressed miRNA (post-normalization) are available compressed in S1 Data.

When inspecting the differential expression results of all normalization methods, the unnormalized data and each dataset separately, there are 33 unique miRNAs that are only shown as significantly (*Q value*<0.05) differentially expressed in ER positive vs ER negative as identified by our normalization method (S2 Fig, S2 Text, list in S6 Table). Contrastingly, other approaches yield far fewer significantly differentially expressed miRNAs. Of the 33 miRNAs uniquely detected by our method, we present four in Table 1 that have fold change greater than 0.15 (absolute $\log_2 > 0.15$, which translates to > 10% change between median expression of ER positive and ER negative tumors).

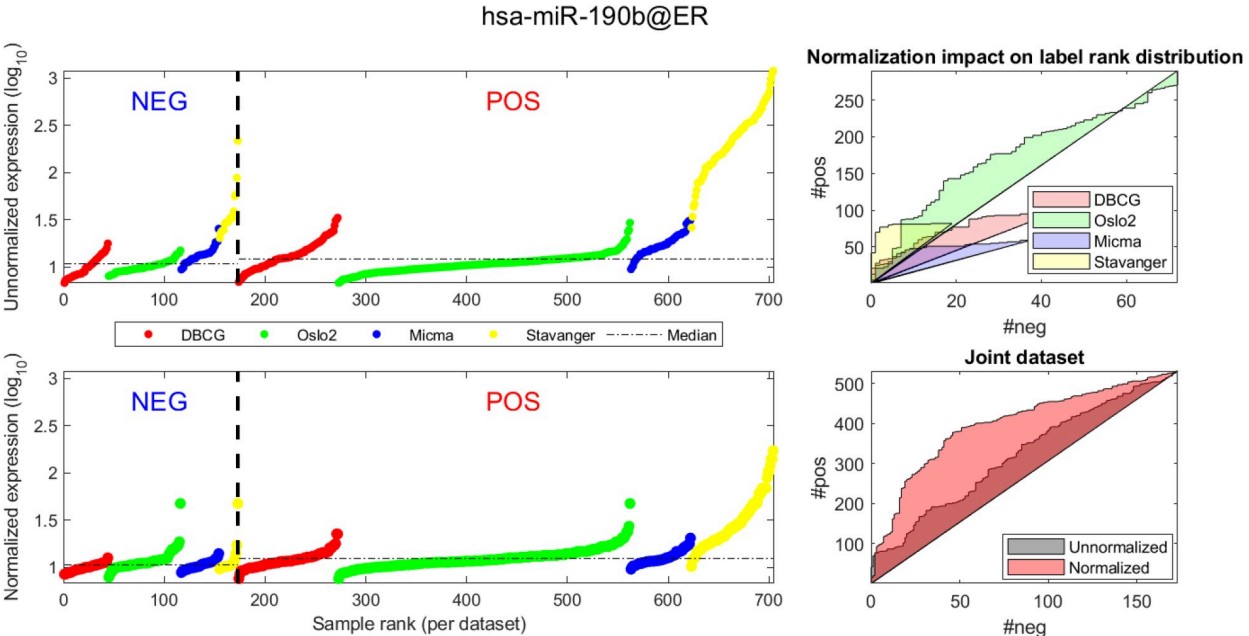

**Fig 4. Visualizing expression of hsa-miR-190b across datasets and samples and in regard to estrogen receptor (ER) positive (pos) vs. negative (neg) differential expression.** (A, D) Expression values (log₂) of each sample before quantile normalization. Samples are ranked by ER status label, then by dataset and finally by ascending expression value. (A, B)-Unnormalized joint dataset. (C, D)-Normalized joint dataset. (B, C) Actual vs expected (via a uniform null model) rank distribution of ER negative (neg) vs positive (pos). Diagonal straight lines bounding a polygon represent a null uniform distribution of positive and negative samples (when ranked by expression value). The colored surface area represents the magnitude of deviation from a uniform distribution. The boundary of the surface is calculated by the cumulative number of ER negative (x axis) vs ER positive (y axis) samples in the ranked (descending) expression vector. Top-illustrating the rank distribution per-dataset (without normalization). Bottom-comparing the joint-dataset distributions when ranking before or after normalization.

To study any breast cancer related functional significance of these top differentially expressed miRNAs we performed miRNA gain-of-function studies in the MCF-7 breast cancer cell line. Here, cell viability was measured as an endpoint after overexpression of the miRNAs. Indeed, one of the miRNAs, hsa-miR-193b-5p, showed a significant reduction in cell viability compared to miRNA negative controls (Fig 6). Furthermore, we looked into data from another functional experiment previously published [32] in the HER2 positive breast cancer cell line KPL4 and here we found that hsa-miR-193b-5p induced apoptosis (as measured by the levels of cleaved PARP), and downregulated the levels of HER2 and phosphorylated ERK upon over-expression. Altogether, these results suggest that miR-193b-5p may exert a tumor-suppressor function in breast cancer, both in an ER+ and a HER2+ context. Interestingly, the other miRNA originating from the same precursor, hsa-miR-193b-3p has been previously shown to directly target ESR1 mRNA and is thus a direct regulator of the ER [33].

Further investigation of the three other top differentially expressed miRNAs shows prior evidence linking them to cancer. For example, hsa-miR-601 is a known prognostic marker and potential tumor-suppressor in breast cancer [34] and hsa-miR-936 was identified as a potential tumor-suppressor miRNA in ovarian cancer [35]. While these findings do not directly validate our findings in ER differential expression, they support the potential association of these miRNA through related mechanisms of cancer pathogenesis.

## Joint analysis with mRNA data

A similar pipeline to the one described in subsection "Dataset pre-processing and coverage" was used to parse the Oslo2 cohort mRNA expression data, using Limma.

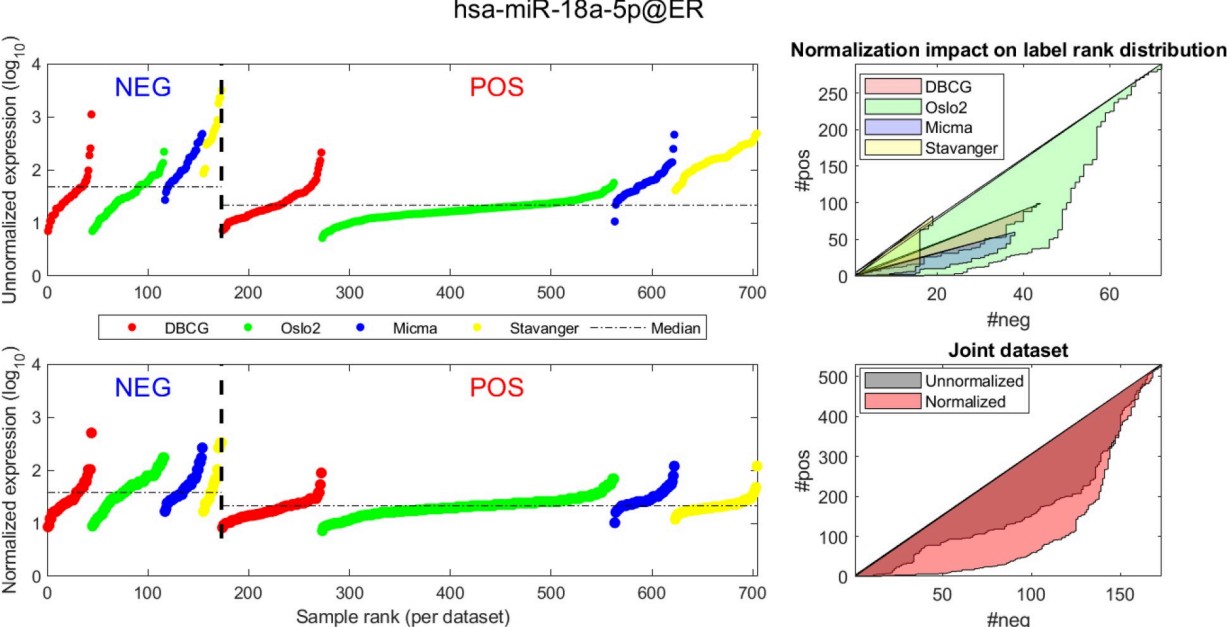

**Fig 5. Visualizing expression of hsa-miR-18a across datasets and samples and in regard to estrogen receptor (ER) positive (pos) vs. negative (neg) differential expression.** Caption description matches the one provided in Fig 4.

We wanted to assess the effect of AQuN normalization on the results of enrichment analysis as performed using both mRNA and miRNA data. To this end we first formed a ranked list of transcripts as follows. For each miRNA, $\mu$, we ranked all mRNAs according to the (ascending) Spearman correlation between the miRNA expression pattern across the entire dataset and the mRNA expression pattern across the entire dataset (paired on matching samples). We denote the resulting ranked gene list, with $\mu$ as a pivot, as $\mathcal{G}_\mu$.

**Effect on gene target enrichment.** For the first analysis we investigated the impact of AQuN normalization on correlations between miRNA and the expression levels of their expected mRNA targets in the Oslo2 dataset. We expect stronger negative correlation after normalization to direct gene targets. To validate this hypothesis, we applied a non-parametric, rank-based analysis using the MiTEA [36, 37] approach. MiTEA is used to evaluate the statistical association between $\mathcal{G}_\mu$ and $\mathcal{C}_\nu$, where $\mathcal{C}_\nu$ is a ranked list of genes wherein the ranking is based on the affinity of the gene as a target candidate for the miRNA $\nu$, taken from TargetScan [38]. A short overview of MiTEA's algorithm is available in the Materials and Methods section.

We declare a matching if MiTEA returns a significant ($\leq$0.001) P-value when $\nu = \mu$. To recapitulate, a matching occurs if the top of two lists of genes overlap to a high degree: the

**Table 1. Top differentially expressed miRNA sorted by fold change.**

| miRNA | Fold Change (log$_2$) | Q-value |
|---|---|---|
| **hsa-miR-601** | -0.18 | 0.048 |
| **hsa-miR-424-3p** | -0.17 | 0.0003 |
| **hsa-miR-936** | -0.15 | 0.027 |
| **hsa-miR-193b-5p** | 0.19 | 0.0002 |

We apply AQuN normalization on the joint dataset and not detected by other approaches. Fold change is defined as $\log_2 \frac{Median\ ER\ positive}{Median\ ER\ negative}$.

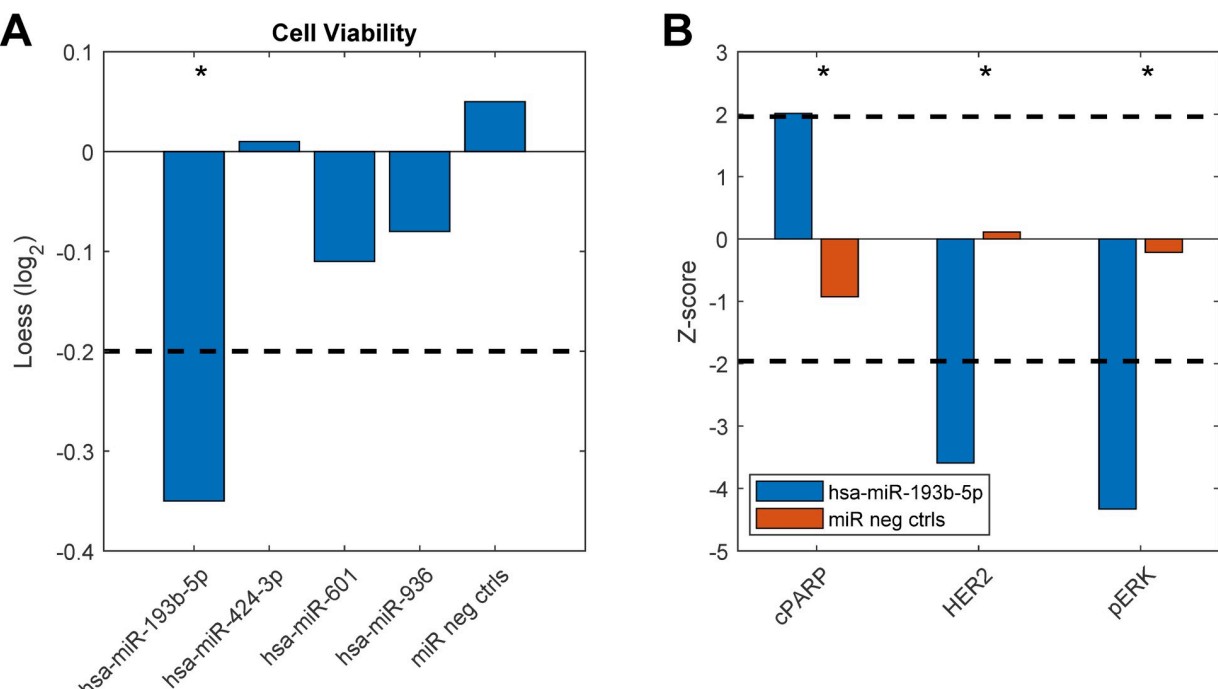

**Fig 6. Functional experiment results.** Breast cancer cell lines were transfected with miRNA mimics (20nM) and assayed for functional effects 72 hours after transfection. A) Cell viability measured in MCF7 breast cancer cells. B) Apoptosis measured by levels of cleaved PARP (cPARP), HER2 and phosphorylated ERK (pERK) protein levels measured in KPL4 cells. The dashed lines indicate cut-off points that were considered significant (see Materials and Methods). Asterisks denote significant effects. Original data from b) are taken from [32].

prominent predicted gene targets (by TargetScan) of miRNA $\mu$ and the list of genes ranked according to their sample-wise anti-correlation with their matched expression levels of miRNA $\mu$. When applying this procedure on a non-normalized miRNA expression we find no matchings. When applying the same procedure on AQuN normalized data we find 6 matchings as detailed in Table 2. For each matched miRNA we also provide supporting evidence of several studies describing its role in breast cancer. We included an extension of this analysis across other datasets and normalization approaches in S2 Table.

We show one such analysis in detail for *hsa-miR-29b* in Figs 7 and 8. Here we follow MiTEA's approach to obtain a statistical assessment of target enrichment for $\mu = \nu = hsa\text{-}miR\text{-}$

**Table 2. Resulting MiTEA matchings on normalized miRNA expression.**

| miRNA | P-value | Q-value | Corroborating studies |
|---|---|---|---|
| hsa-miR-29b | 1.28E-08 | 1.73E-06 | [39–41] |
| hsa-miR-106b | 1.96E-06 | 1.11E-04 | [42–44] |
| hsa-miR-200b | 1.06E-04 | 5.54E-03 | [45–47] |
| hsa-miR-30d | 4.38E-04 | 1.19E-02 | [48, 49] |
| hsa-miR-96 | 9.02E-05 | 1.53E-02 | [50, 51] |
| hsa-miR-182 | 4.58E-04 | 4.43E-02 | [52, 53] |

P and Q values are color coded by magnitude where from green (more significant results) to red (less significant results). None of these statistically significant associations between pivot miRNAs and their targets is observed when using the raw, un-normalized data. Nor is any other matching miRNA target enrichment observed in the unnormalized data.

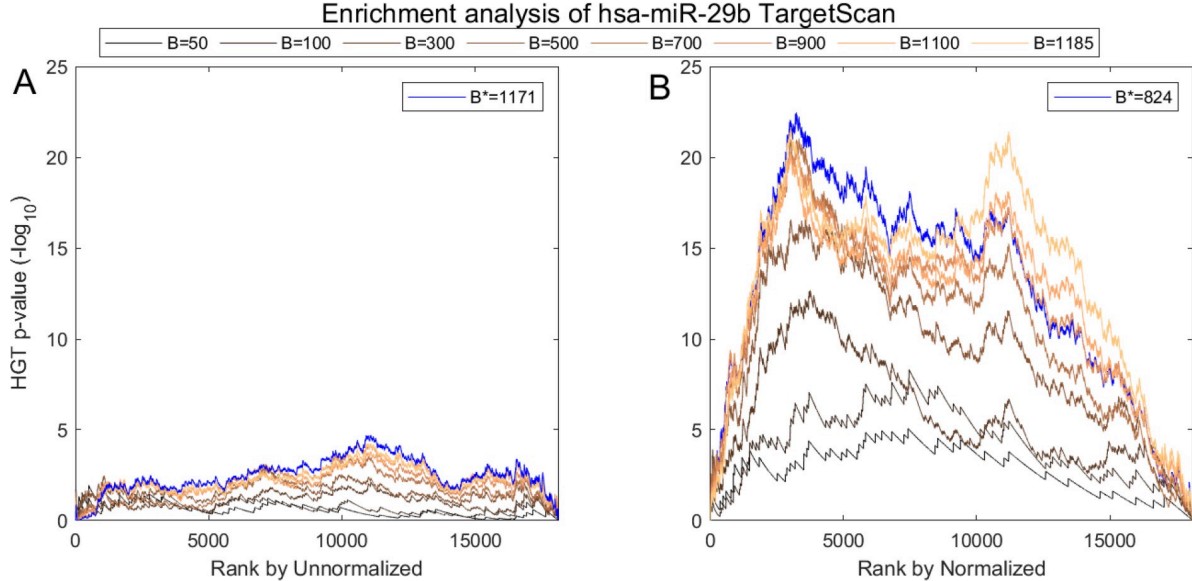

**Fig 7. Impact of normalization on the correlation between hsa-miR-29b expression and its in-silico predicted targets according to TargetScan.** (A) AQuN normalized vs Unnormalized (B) miRNA showing normalized is more negatively correlated to the prominent hsa-miR-29b targets in TargetScan as evident in stronger enrichment values.

29b and $B = \{1, \ldots, |\mathcal{C}_v|\}$ binary vectors $\mathcal{B}(\mu, v, B)$. We present the results on various $B$s and the optimal $B^*$ for both unnormalized and normalized miRNA expression.

**Effect on gene ontology (GO) enrichment.** We applied GOrilla [36] to identify gene ontology enrichment in $\mathcal{G}_\mu$ on both unnormalized miRNA expression and on normalized miRNA expression. Given a ranked list $\mathcal{G}_\mu$, GOrilla produces a binary vector $\mathcal{B}(\mathcal{G}_\mu, \omega)$ for each gene ontology term, $\omega$, in which a gene is labeled as binary '1' if it belongs to $\omega$. Next, GOrilla computes mHG p-values, correcting them across GO terms. Fig 9 is a scatterplot comparing between our results on unnormalized and normalized hsa-miR-29b lists. The findings from this analysis are in line with previous studies that have linked the miR-29 family with tumor growth and metastasis [40, 54, 55].

## Discussion

We have presented an integrative analysis technique and applied it to jointly analyze human breast cancer miRNA expression datasets spanning different studies and utilizing different measurement technologies. Our approach is powerful in its ability to increase statistical power without apparent adverse effects on precision, as exemplified by several downstream analysis results. Our normalization method (AQuN) is based on a slight adaptation to standard (a.k.a. vanilla) quantile normalization. Vanilla quantile normalization averages values across samples with the same rank, while our method averages values across samples within the same percentiles (computed per sample). This has the effect of lowering the impact of within-quantile noise when computing rank-based statistics. Additionally, our method, as defined, can support normalization of multiple cohorts that contain only partial overlaps in their evaluated miRNAs. Correctly applying AQuN requires a basic understanding of the impact it has on downstream statistics. In this work we focused on applying nonparametric rank-based statistics to downstream analyses. Our normalization approach can apply to parametric analyses as well. Further discussing parametric analysis is out of scope for this work. We offer a short discussion

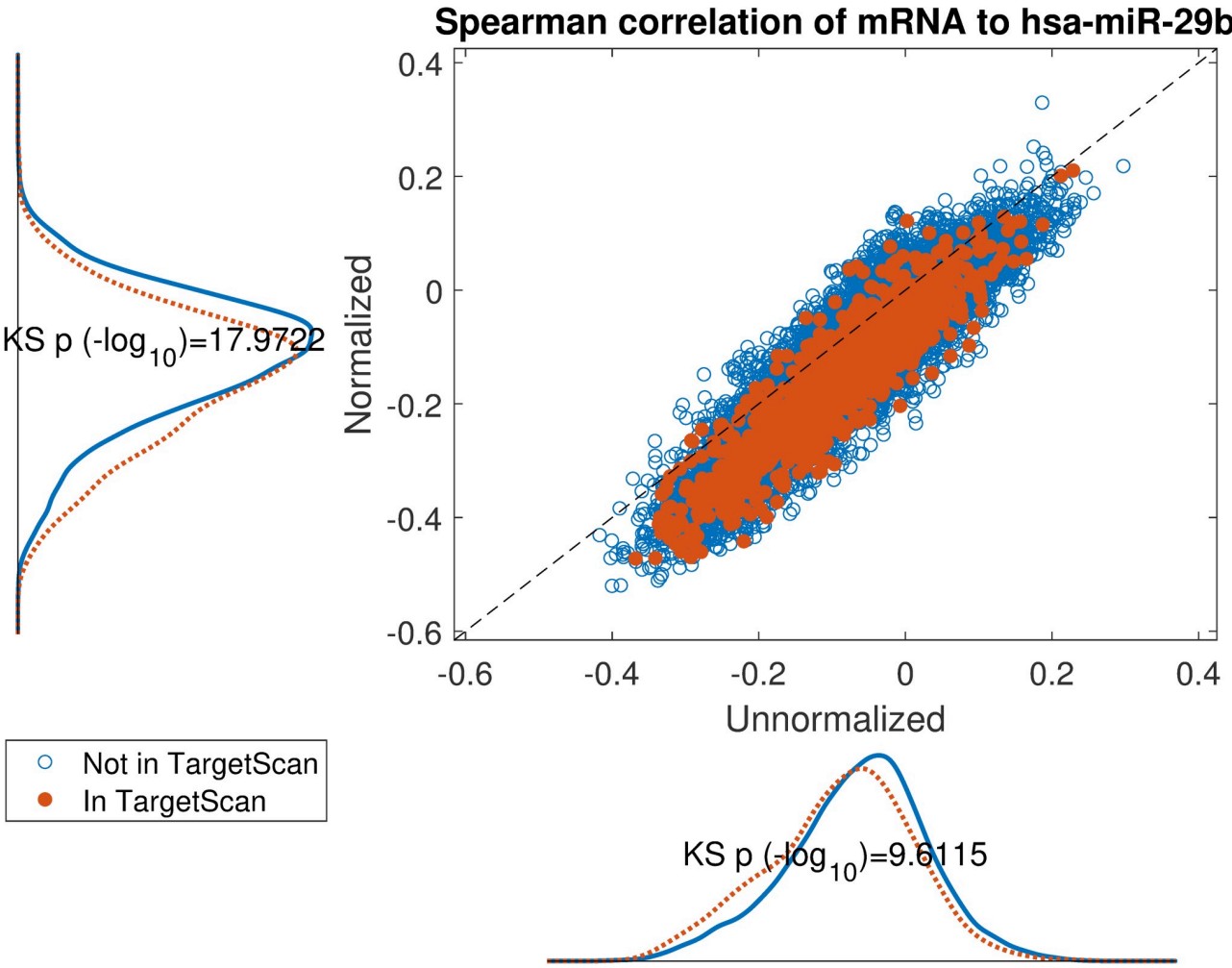

**Fig 8. Impact of normalization on the correlation between hsa-miR-29b expression and its in-silico predicted targets according to TargetScan.**
Scatter plot of spearman correlation on normalized miRNA or unnormalized miRNA expression. If the target mRNA appears in TargetScan it is highlighted in orange. The marginal distributions are shown parallel to the axes and corresponding Kolmogorov-Smirnov test p-values display an overall lowered correlation for TargetScan candidates on normalized data.

on the impact of normalization on intra-sample rankings and intra-miRNA rankings (see S3 Text and S3 Fig).

The first point to address, in terms of impact on downstream statistics is in the context of differential expression. We focus the discussion on ER related differential expression. When comparing the normalized joint dataset with per-dataset analyses we observe stronger p-values, yielding more statistically significant candidates after applying multiple hypothesis correction procedures. In Fig 3, we illustrate this result through a shift in the cumulative distribution of Wilcoxon Rank-sum FDR corrected Q-values calculated for the differential expression of ER positive and negative samples. In Fig 10 we present a per dataset drill-down in to the analysis presented in Fig 2. For some miRNAs, we observe a tradeoff between higher absolute fold-change and higher rank-sum $-\log_{10}$ Q-values. For example note hsa-miR-135b that has $> -8\times$ fold change for Stavanger, but at a fairly low $-\log_{10}$ Q-value $< 4$ while after joint analysis it demonstrates only $> -2\times$ fold change but at $-\log_{10}$ Q-value $> 18$.

A second important point is the increase in statistical power that is afforded through the integration of several datasets. One of the main motivating reasons for jointly analyzing

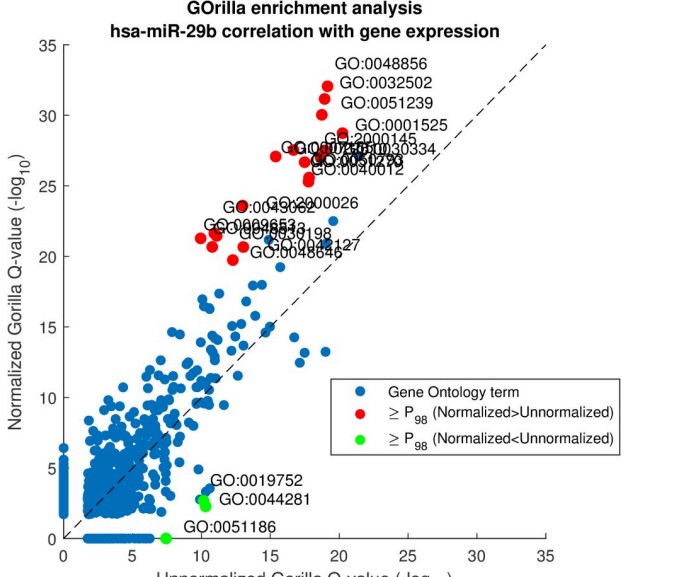

**Fig 9. GOrilla enrichment analysis comparison of hsa-miR-29b correlation with gene expression before and after miRNA normalization with AQuN.** Showing scatter of GO term Q-values before and after AQuN. Red dots depicted with "≥P98" are above the 98th percentile of Normalized–Unnormalized Q-values (-log₁₀) and green dots are for Unnormalized–Normalized. Right side panel shows a list of GO terms in the ≥P98 group.

datasets collected in different places, times and possibly using different measurement technologies is the fact that the combined dataset supports higher statistical power. As we have shown in Fig 3, this increase of power is not attainable when naively joining the dataset or when normalizing with the presented alternatives.

We present a theoretical statistical a-priori power analysis [56] to put in context the advantage of jointly analyzing the datasets investigated in the current work. Remember that power is used in statistics to quantify the recall of a statistical test, i.e. the probability of correctly rejecting the null hypothesis. The test evaluated in this analysis is Wilcoxon rank-sum as applied for our differential expression analysis in the results section under subsection "Differential expression reveals novel breast-cancer associated miRNA". Power is only meaningful in the context of an expected effect size (measured herein using Cohen's d [57]), as larger differences and less variance in samples implies a smaller sample size is required to decide there is a difference between two populations. For the purpose of this analysis we assume allocation ratio = 1 (i.e. equal group sizes), while in the ER examples shown in Fig 11 actual ratios of Negative vs Positive ER samples are 0.44, 0.24, 0.63 0.23 and 0.32 for DBCG, Oslo2, Micma, Stavanger and Joint, accordingly–further reducing expected power. We overlay the theoretical plot with empirical effect sizes measured per dataset in hsa-miR-29b-3p and has-miR-18a-5p which we have identified as miRNAs of interest in Table 2 and Figs 4 and 5, accordingly.

A potential line of inquiry to follow up on from this study is to compare AQuN results on other sets of cohorts and with other normalization approaches. We have applied a preliminary analysis on a second set of cohorts, consisting of the TCGA [58] and Tahiri [14] cohorts. We performed a differential expression analysis comparing cancer and control samples and include a 5th normalization method previously referenced which is relevant in the context of case-vs-control experimental setups. Our results are presented in S4 Fig. When sorting miRNA according to their differential expression -log₁₀(Q-value), we observe many known oncomiRs [59] are ranked higher and have more significant Q-values (S1 Table).

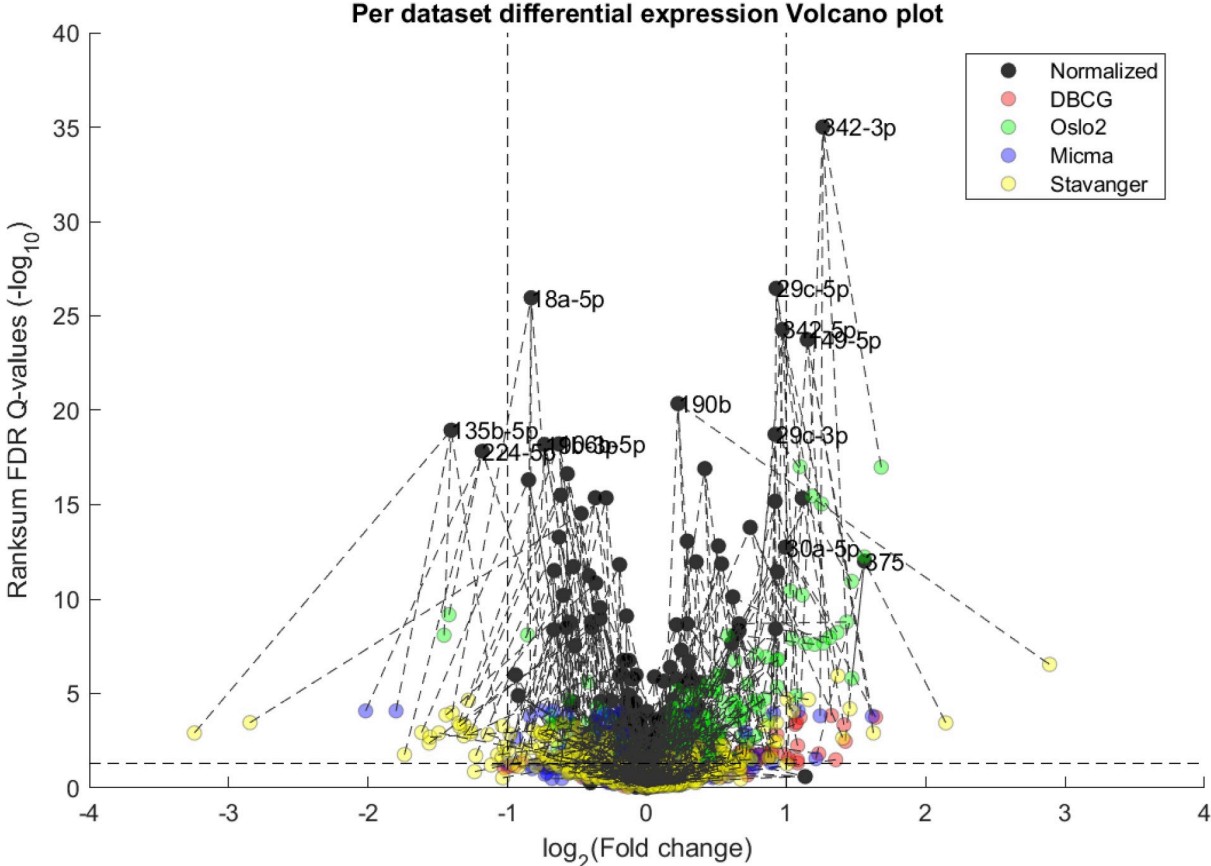

**Fig 10. Volcano plot of per-dataset Differential Expression of ER positive vs ER negative from Fig 1.** Here we include a both joint normalized and per dataset results. We observe an overall increase in statistical significance as dark points tend to be higher on the y-axis than their corresponding-colored points (indicated by dashed lines), as would be expected from the increase in statistical power. In some miRNA this can come at the cost of a lower detected fold-change as compared to some individual datasets.

Overall, we provide multiple lines of evidence supporting the joint analysis of miRNA expression using nonparametric statistics. Our analysis yields potential novel biomarkers as exemplified by hsa-miR-193b-5p and its potential tumor-suppressor role in breast cancer. While these results require further validation, we demonstrate how stronger statistical evidence can be obtained in suggesting candidates and in prioritizing follow-up studies.

## Materials and methods

### Ethics statement

The Stavanger cohort was approved by REC Region West, approval number 2010/2014. By this approval, none of the patients were required to provide written informed consent to participate. All insights in a patient's journal were monitored electronically, and all except the treating physician were required to state the reason why they needed to read that patient's journal. This log was always open for the patient to view.

### Overview

We used miRNA expression data from three previously published breast cancer datasets along with a newly released, fourth, miRNA dataset. These datasets were acquired from fresh-frozen

**Fig 11. Statistical power as a function of sample size and expected effect size (measured in Cohen's d [66]).** Dotted line plots illustrate an a-priori power analysis for one-tailed Wilcoxon Rank Sum (WRS) test for different effect sizes. Overlaid in squares and triangles are effect sizes, d, for the differential expression of hsa-miR-18a-5p and hsa-miR-29b-3p, accordingly, in ER positive vs ER negative samples as estimated empirically over the joint dataset on non-normalized data. Power values are estimated via (linear, 2D) interpolation on different dataset sizes.

material with different minimal number of tumor cells criteria, using different technologies and experimental protocols as overviewed in **Table 3**. In addition, we utilized mRNA expression to further investigate the effect of normalization using one of the cohorts. We examine miRNA normalization also in the context of jointly analyzing these measurements. Below we elaborate our considerations in the selections made during the normalization process and our means of providing evidence for validating these results.

**Table 3. Technical details of platforms used for expression measurements for the four different cohorts.**

| Color code | Dataset | Manufacturer | Technology | Version | Accession number | Number of samples |
|---|---|---|---|---|---|---|
| | DBCG[25]–miRNA | Agilent | Human miRNA Microarray Kit | (V2 G4470B) design id 019118 | GSE46934 | 149 |
| | Oslo2[15]–miRNA | Agilent | Human miRNA Microarray Kit (V2) | v14 Rev.2 design id 029297 | GSE81000 | 425 |
| | Oslo2[15]–mRNA | Agilent | SurePrint G3 Human GE 8x60K Microarray | (Probe Name Version) 028004 | GSE80999 | 381 |
| | Micma[11]–miRNA | Agilent | Human miRNA Microarray Kit | (V2 G4470B) design id 019118 | GSE19536 | 101 |
| | Stavanger–miRNA | Exiqon | miRCURY LNA Array | v.11.0 | | 109 |

Datasets are color coded consistently throughout the paper. miRNA expression colors are highlighted compared to mRNA measurements.

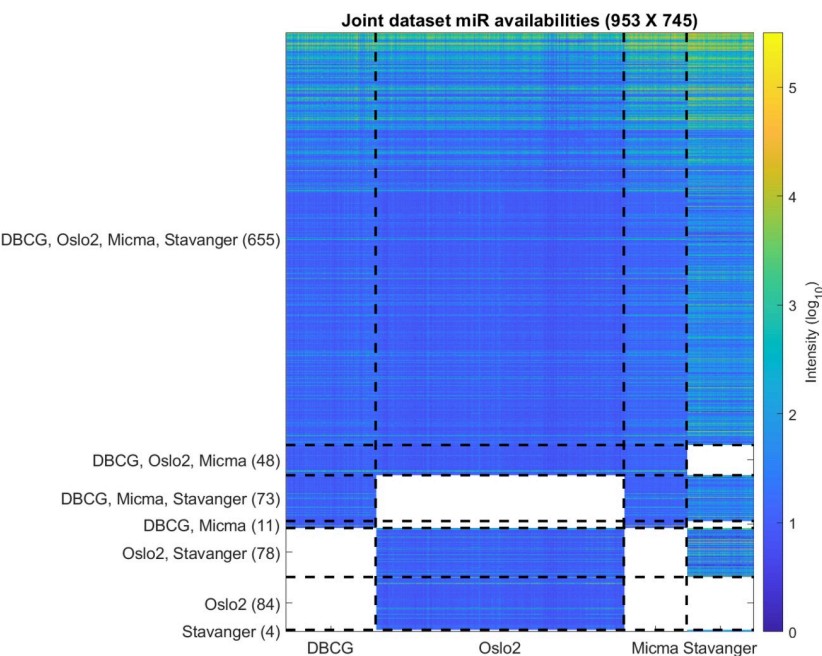

**Fig 12. Overview of the miRNA coverage in the dataset.** Each row represents one miRNA. Each entry represents the intensity ($\log_{10}$) in a specific sample. Dashed vertical lines separate between samples from the four datasets. Dashed horizontal lines separate between groups of miRNAs by their dataset availability. Blank (white) entries correspond to miRNAs that are missing from a dataset.

## Dataset pre-processing and coverage

Each miRNA dataset is read from a single-channel image analysis output file acquired from their corresponding GEO repositories (referenced in Table 3) and preprocessed in R using the Limma [60] package. We note that while Stavanger (Exiqon) data contains a pooled-reference second channel, this measurement is not utilized in our analysis (further discussed in S1 Text). Initially, control probes are removed, and the data is corrected by background intensity normalization [61]. Same-probe replicate samples are replaced by their median value. Probe ids are mapped to their corresponding miRbase v22 accession using miRBaseConverter [62]. Missing or deleted accession IDs are discarded. Multiple probes that map to the same miRNAs are again replaced by their median value. Next, we apply arrayQualityMetrics [63] (resulting Quality Control reports are available compressed in S2 and S3 Data) and filter out samples that are marked as outliers by all three outlier detection criteria ($L_1$-Distance between arrays, Boxplot, MA plot). We thereby filtered out 6, 30, 12 and 2 outliers from DBCG, Oslo2, Micma and Stavanger, respectively. Next, we apply minimum subtraction to avoid log scaling issues with negative numbers where applicable. The joint dataset table is then compiled by applying a "full outer-join" relational operation on the miRbase accession IDs as key. The resulting miRNA cross-dataset table is visualized in Fig 12 (and available in S3 Table as raw data and S4 Table as normalized data with corresponding clinical labels in S5 Table).

**Batch effects in joint data.** We tested for rank-order consistency of miRNA expression in pairs of datasets (Fig 13). For each miRNA we take the median of its expression, or similarly, intra-sample percentile, across all samples belonging to the same dataset. We display the resulting values for each pair of datasets in a scatterplot matrix considering the miRNAs (n = 655) present in all four cohorts. This analysis shows that the Stavanger data appears to

## Pairwise dataset miRNA correlations

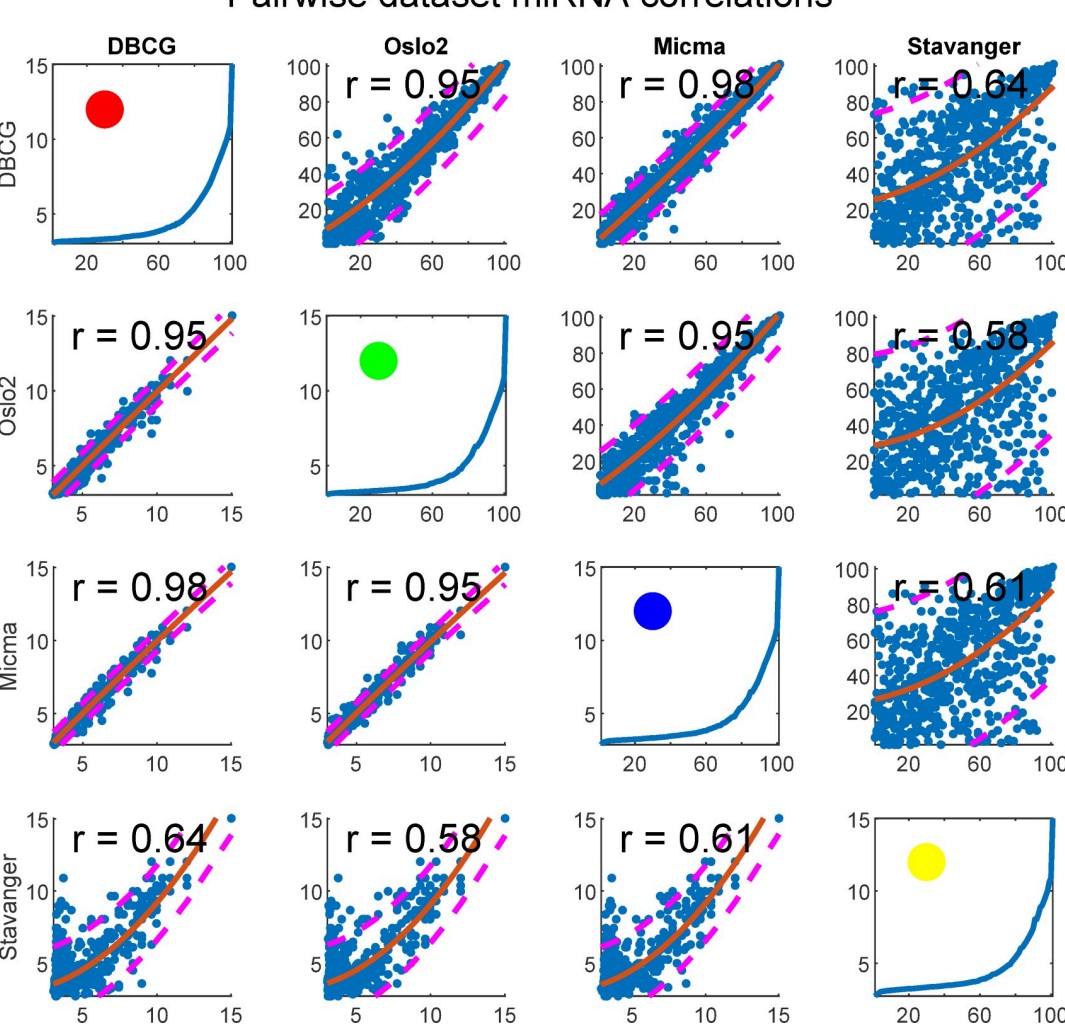

**Fig 13. Scatterplot matrix of quantile normalized data showing miRNA expression reproducibility across dataset pairs.**
Each subplot depicts a pair of datasets. In the upper-diagonal-subplots, each point corresponds to a single miRNA's median (across samples) rank (intra-sample) in each dataset. Similarly, the bottom-diagonal shows median log2 expressions in place of ranks. A second degree polynomial curve is fitted and prediction intervals at confidence level 0.8 are plotted as dashed lines. Spearman correlation is given for each subplot. Figures at the diagonal show percentile plotted against expression and a circle represents the dataset colorcode as related to other figures in the paper.

behave differently, presumably due to its fundamentally different measurement technology (Exiqon LNA—Locked Nucleic Acids vs Agilent Microarray).

We further visualize the batch-clustering behavior of the unnormalized joint dataset in Fig 14. On the left panel (A) we present hierarchical clustering of the data. Edges of sub-trees in the dendrogram are color-coded by the dataset when all leaves in the subtree belong to samples from the same original dataset. We observe a visual clustering of colors, especially evident for yellow (Stavanger) being clustered as an outgroup. In the middle panel (B) we show a silhouette plot, depicting the clustering consistency according to dataset. We observe a substantial portion of samples that are well assigned to their cluster with large silhouette values, and only a small portion are mis-assigned, again showcasing how batch effects dominate sample pairwise-distance pattern behavior. Finally, on the right panel (C) we present a visualization of the

Visualizing sample-wise batch effect in the joint dataset

**Fig 14. Visualizing batch effects in the combined cross-tech miRNA dataset considering the unnormalized data.** (A) Dendrogram with edges colored by dataset. Note that the tree root is outside the displayed axis range. (B) Silhouette plot [67] showing that most samples cluster according to the dataset they originate from. (C) Pairwise Euclidean distances showing a block structure that agrees with the sample dataset of origin.

sample-wise pairwise Euclidean distance matrix with dashed lines separating between samples of the same dataset. The block-diagonal structure that evidently results from coloring according to distances corresponds well to the dashed lines separating samples from different datasets. This analysis demonstrates the prevalence of batch effects in the joint datasets.

## Adjusted quantile normalization (AQuN)

In this section we describe our quantile-normalization-based strategy for analyzing combined cross-technology miRNA datasets.

Let $M$ be a batch collected, joint dataset. $M \in R^{n \times m}$ where $M(i,j)$ is the log measured intensity value of miRNA $i$ in sample $j$. We note that the $j$-th column of $M$, denoted $M(:,j)$, corresponds to the $j$-th sample and that the $i$-th row, $M(i,:)$ corresponds to the $i$-th miRNA in the joint dataset.

Define $\beta(M(:,j)) = k$ as the experiment batch id during which sample $j$ was collected.

We note the following distinction between missing values in $M$:

$$M(i,j) = \begin{cases} nan & \text{miRNA } i \text{ was not sampled in } \beta(M(:,j)) \\ 0 & \text{miRNA } i \text{ was sampled in } \beta(M(:,j)) \text{ but not detected in sample } j \\ \geq 0 & \text{otherwise} \end{cases}$$

Let $MFP(i,j) = 1$ if miRNA $i$ is missing from platform $\beta(M(:,j))$ and $MFP(i,j) = 0$ otherwise (indicates if $i$ is missing in the platform $j$ was measured in).

| | Adjusted Quantile Normalization ($M$): | |
|---|---|---|
| 1. | $\mathcal{D} \leftarrow M + N(0, \epsilon)$ | Jitter $M$ to break rank ties. |
| 2. | Let $P(i,j)$ = the percentile of $\mathcal{D}(i,j)$ within $\mathcal{D}(:,j)$. | *nan*s are ignored in percentile computation. Note: $P(i,j) \in$ [0,100] |
| 3. | $Q(i,j) = \underset{1 \leq t \leq m}{median} \{\mathcal{D}(s,t) : P(s,t) = p(i,j)\}$ | Transforms values to the cross-sample-median of the corresponding per-sample-quantile. |
| 4. | $Q(i,j) = nan$ if $MFP(i,j) = 1$ | |

A description of this process in words is that it replaces all present expression values with the corresponding median value of all samples within the same percentile. The underlying assumption is that a measured expression is volatile due to technical differences and measurement noise, however, (sample-based) percentiles are assumed to be stable up to the biological differences between samples. In addition—platform coverage differences are addressed.

The overall impact of applying AQuN to the distribution of expression values and to quantified batch effects as measured by the silhouette coefficient is further presented in S1 Fig.

### Functional experiments

Functional experiments were performed as previously described [32, 33] with the breast cancer cell lines MCF7 and KPL-4. The lysate microarray data measuring apoptosis in the form of cleaved PARP (cPARP), HER2 and phosphorylated ERK (pERK) protein levels after 72 hours were previously published (data taken from S2 Table of referenced paper and provided as S7 Table herein) [32]. Values ±2 × standard deviation (SD) were considered as significant, which corresponded to a threshold of |1.96|. For the cell viability data, MCF7 cells were transfected with the Dharmacon miRIDIAN microRNA mimic library v.10.1 (20 nM) and incubated for 72 hours. The cell viability was measured with CellTiter-Glo assay (Promega Corp, Madison, WI, USA) according to manufacturer's protocol. The experiments were done with two biological replicates. The data were normalized by a Loess method [64] and log2-transformed. Values ±2 × SD, were considered as significant, which corresponded to a threshold of |0.2|. In both experiments the average of two different miRNA mimic controls from two replicates was used as negative controls (miRIDIAN microRNA Mimic Negative Control #1 from Dharmacon and pre-miR negative control #2 from Ambion). The transfection efficiency of miRNA mimics has been determined previously [33].

### MiTEA algorithm overview

Briefly, for each prefix $\Pi_B(\mathcal{C}_v)$ of $B$ most-prominent candidate targets in $\mathcal{C}_v$, MiTEA produces a binary vector, $\mathcal{B}(\mu, v, B)$, such that, $g_i$, the $i$-th gene in $\mathcal{G}_\mu$ is labeled "1" if and only if it is in the candidate prefix, i.e. $g_i \in \Pi_B(\mathcal{C}_v)$. MiTEA then computes an approximate minimum hypergeometric (mHG [36, 57]) P-value to quantify whether the $B$ proposed targets are enriched at the top of the $\mathcal{G}_\mu$ list or not. Finally–MITEA applies an FDR correction (using the Benjamini-Hochberg procedure [65]) across evaluated $v$s and reports the set of miRNAs associated with the ranked target list $\mathcal{G}_\mu$ and their associated Q-values.

### Supporting information

**S1 Data. A zip file containing figures for the top 40 differentially expressed miRNA (post-normalization) on ER clinical label.** Contains additional figures per miRNA pertaining to the analysis presented in Fig 2.
(ZIP)

**S2 Data. A zip file containing quality control reports for miRNA datasets.** Generated by the arrayQualityMetrics package, as described in "Dataset pre-processing and coverage". Open index.html in either folder to view the detailed report data.
(ZIP)

**S3 Data. A zip file containing quality control reports for mRNA datasets.** Generated by the arrayQualityMetrics package, as described in "Dataset pre-processing and coverage". Open

index.html in either folder to view the detailed report data.
(ZIP)

**S1 Fig.** Top) Kernel density estimates of each sample colored by their corresponding dataset. The resulting normalized distribution is overlaid in black. Bottom) Impact of normalization on per-sample silhouette coefficient measured for clustering by dataset. 602/745 samples have lower silhouette coefficients after normalization in comparison to before normalization, demonstrating an overall alleviation of batch effect per dataset. Marginal distributions are shown to highlight differences between datasets.
(TIF)

**S2 Fig. Venn diagram comparing the number of differentially expressed miRNAs surfaced by different normalization approaches.** We observe a larger set of unique miRNAs detected by our normalization approach compared to other approaches.
(TIF)

**S3 Fig. Histograms of Sample-wise and miRNA-wise Spearman correlation coefficient ($\rho$) between expression before and after normalization.**
(TIF)

**S4 Fig. Differential expression analysis on TCGA, Tahiri cohorts (Cancer vs. Normal tissue).** The results shown here follow the same analysis described in Fig 3 in the manuscript. We note the additional evaluated method "Percentile Normalization" was added as it is only relevant in a case-vs-control setup, as evaluated here. Note that the "Percentile Normalization" curve is overlaid by the random permutation curves (dashed black curves).
(TIF)

**S1 Table.** A set of known oncomirs (Wikipedia) their Q-value (cells conditionally formatted from green to red) and ranks (cells conditionally formatted from blue to red) when sorted according to the differential expression Q-value from the analysis shown in S4 Fig. We observe a higher set of oncomirs return as differentially expressed, and many are more highly ranked (including very prominent ones such as miR-21, miR-18a).
(XLSX)

**S2 Table. An extension of the analysis presented in Table 2 which includes a comparison with an additional normalization method (Vanilla quantile) and across two more datasets (Micma and Stavanger).**
(XLSX)

**S3 Table. Joint dataset table raw measurement data as depicted in Fig 12.**
(XLSX)

**S4 Table. Joint dataset table from S3 Table, normalized by AQuN.**
(XLSX)

**S5 Table. Corresponding clinical labels for S3 Table and S4 Table.**
(XLSX)

**S6 Table. A list of 33 unique differentially expressed miRNAs between ER positive vs ER negative as identified by our normalization method.**
(XLSX)

**S7 Table. Functional experiment results table presented as S2 Table published in Leivonen S-K et al. [32].**
(XLSX)

**S1 Text. Discussion on joint one-color and two-color analysis of available Stavanger data.**
(DOCX)

**S2 Text. Caption for S2 Fig–Venn diagram of differential expression results.**
(DOCX)

**S3 Text. Caption for S3 Fig and discussion on AQuN impact on data rankings.**
(DOCX)

## Acknowledgments

We thank the Yakhini research group for important insights and comments throughout the research process. We thank Prof. Benny Chor for his advice and encouragement for pursuing this work.

The results published here are in whole or part based upon data generated by the TCGA Research Network: https://www.cancer.gov/tcga.

MR Aure was a postdoctoral fellow of the South Eastern Norway Health Authority (https://www.helse-sorost.no/south-eastern-norway-regional-health-authority) under grant 2014021 to Anne-Lise Børresen-Dale and a research fellow of the Norwegian Cancer Society (https://kreftforeningen.no/en/about-us/) under grant 711164 to Vessela N Kristensen.

The Stavanger database was sponsored by grants from the Folke Hermansen Foundation in 2007, a grant provided by the Stavanger University Hospital research fund in 2009 and a PhD-fellowship from Helse Vest in 2009.

Folke Hermansen Foundation (http://www.folke-fondet.org/) grant 2007 entitled "Early breast cancer prognostication by genomics and proteomics", EAMJ received the unnumbered grant.

Stavanger University Hospital, research department (https://helse-stavanger.no/en) grant entitled "Comparing the prognostic and predictive value of microRNA, gene expression signatures and proliferation in early node negative breast cancer", EAMJ received the unnumbered grant.

## Author Contributions

**Conceptualization:** Shay Ben-Elazar, Miriam Ragle Aure, Suvi-Katri Leivonen, Ole Christian Lingjærde.

**Data curation:** Miriam Ragle Aure, Kristin Jonsdottir, Suvi-Katri Leivonen, Kristine Kleivi Sahlberg.

**Formal analysis:** Shay Ben-Elazar, Miriam Ragle Aure, Zohar Yakhini.

**Funding acquisition:** Miriam Ragle Aure, Vessela N. Kristensen, Emiel A. M. Janssen, Zohar Yakhini.

**Investigation:** Shay Ben-Elazar, Miriam Ragle Aure, Kristin Jonsdottir, Suvi-Katri Leivonen, Kristine Kleivi Sahlberg, Zohar Yakhini.

**Methodology:** Shay Ben-Elazar, Miriam Ragle Aure, Ole Christian Lingjærde, Zohar Yakhini.

**Project administration:** Shay Ben-Elazar, Zohar Yakhini.

**Resources:** Vessela N. Kristensen, Emiel A. M. Janssen, Kristine Kleivi Sahlberg, Zohar Yakhini.

**Software:** Shay Ben-Elazar.

**Supervision:** Vessela N. Kristensen, Emiel A. M. Janssen, Kristine Kleivi Sahlberg, Ole Christian Lingjærde, Zohar Yakhini.

**Validation:** Shay Ben-Elazar, Miriam Ragle Aure, Suvi-Katri Leivonen, Zohar Yakhini.

**Visualization:** Shay Ben-Elazar.

**Writing – original draft:** Shay Ben-Elazar, Miriam Ragle Aure, Zohar Yakhini.

**Writing – review & editing:** Shay Ben-Elazar, Miriam Ragle Aure, Kristin Jonsdottir, Suvi-Katri Leivonen, Vessela N. Kristensen, Emiel A. M. Janssen, Kristine Kleivi Sahlberg, Ole Christian Lingjærde, Zohar Yakhini.

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
