## [Decision Letter · Decision Letter 0]

8 Aug 2020

Dear Dr. Ben-Elazar,

Thank you very much for submitting your manuscript "miRNA normalization enables joint analysis of several datasets to increase sensitivity and to reveal novel miRNAs differentially expressed in breast cancer" for consideration at PLOS Computational Biology.

As with all papers reviewed by the journal, your manuscript was reviewed by members of the editorial board and by several independent reviewers. In light of the reviews (below this email), we would like to invite the resubmission of a significantly-revised version that takes into account the reviewers' comments.

We cannot make any decision about publication until we have seen the revised manuscript and your response to the reviewers' comments. Your revised manuscript is also likely to be sent to reviewers for further evaluation.

Sincerely,

Denis Thieffry, PhD

Associate Editor

PLOS Computational Biology

William Noble

Deputy Editor

PLOS Computational Biology

Reviewer's Responses to Questions

**Comments to the Authors:**

**Reviewer #1: **

In the manuscript (ID PCOMPBIOL-D-20-00931), authors present a non parametric normalization method, called AQuN for Adjusted Quantile Normalization, to combine miRNA expression data from multiple breast cancer datasets into a joint dataset. They detect 33 miRNAs to be differentially expressed in estrogen receptor (ER) positive versus ER negative samples that do not emerge as statistically significant when separately analyzing these datasets. In particular, they perform miRNA mimics assays on breast cancer cell lines to validate the effect of has-miR-193b-5p on cell viability and apoptosis.

Although the article deals with a relevant topic, such as integrative analysis of microRNA expression data, it does contain major issues that require to be addressed:

1) The authors evaluate the impact of the AQuN normalization on batch effect correction based on more significant results obtained in differential expression analysis (lower Q-values) for most miRNAs. This is not conclusive to determine to what extent the impact of normalization is beneficial. It would be more appropriate to apply a test metric for batch correction such as adjusted rand index (ARI) or others to quantify the extent to which AQuN normalization removes batch effects while increasing subtyping accuracy.

2) Performances of the proposed AQuN normalization method should be more extensively compared with other existing methods, including the non-parametric approach cited in [22], in terms of their efficacy and efficiency to integrate batches while maintaining tumor subtypes separation.

3) Figure 2 is difficult to interpret and does not clarify how the impact of normalization on single miRNAs is evaluated.

4) In the miRNA mimics assay, it would be worthy showing the transfection efficiency of hsa-mir-193b by qRT-PCR or other techniques.

Minor comments

5) A brief description of the AQuN method should be included in the results section, as it is an innovative method and its implementation is a core part of the work.

6) Figure 1 (top panel) and Fig 6 are difficult to read, it is not clear what the dashed lines that join the points correspond to. Volcano plots with dashed lines are not the ideal visualization to show an overall increase of statistical significance.

7) Table 1 reports data on the top four miRNAs identified by differential expression analysis. Results about all the 33 differentially expressed miRNAs can be report as supplementary material.

**Reviewer #2: **

Summary:

The authors developed a miRNA joint analysis approach based on applying adjusted quantile normalization (AQuN) across samples from various sources. In the merged breast cancer dataset using their approach, they discovered and validated that over-expression of has-miR-193b-5p was a new signature of the estrogen positive (ER) group. In short, this approach aims at leveraging statistical power of publicly available miRNA assays to identify novel biomarkers and preliminary therapeutic targets.

Major Comments:

The main concern of this approach development paper is that the authors haven’t done a comprehensive comparison between the outcomes of their proposed method and the original results in the studies they cited using the same dataset. And some results using their new approach is confusingly contradicted. Suggestions for improvements are listed below:

1. From the unnormalized and merged dataset in this paper, hsa-miR-190b does not look like a marker between ER+ and ER- group which is referred in Reference [30]. To evaluate the performance of proposed AQuN, the authors should have applied AQuN on the same dataset as that has been used in Reference [30]. Same for the has-miR-18a-5p.

2. The authors identified four miRNAs with log2-fold-change above 10% and FDR less than 0.05 using AQuN on the merged dataset. However, only hsa-miR-193b-5p had indeed changed the viability of cells in the functional validation. The authors should clarify the following questions:

a) The authors said they transduced miRNA mimics into MCF7 cells to validate hsa-miR-601, 424-3p, 936 and 193-5p. Considering three out of four of these miRNAs were identified as significantly downregulated in ER- compared to ER+, isn’t that a KO, instead of transduction, required for each of the three miRNAs?

b) How much hsa-miR-193b-5p was overexpressed in the MCF7 cells in the validation? Was it close to 10%, or much higher than that? If the latter, it might be explained as a cell burden to be toxic, instead of due to the real impact from this miRNA.

c) In Reference [34], hsa-miR-601 was downregulated in breast cancer tissue compared to the control. The authors should have test if their method could confirm this using a case-vs-control setup, instead of ER-positive-vs-ER-negative. The reference is not a support of finding hsa-miR-601 differentially expressed between ER+ and ER-, making it hard to judge the robustness of AQuN. The for hsa-miR-936.

d) how hsa-miR-936, a tumor-suppressor in ovarian cancer, indicates its role in breast cancer? And it was identified highly expressed in the ER- group by the author’s method again.

3. If I understand correctly, there are totally 655 miRNAs to be studies in the merged dataset. Using MiTEA, the authors declared that no matching between mRNAs and these miRNAs was found when the dataset was unnormalized while 6 matchings between mRNAs and these miRNAs were found after AQuN. That only accounts for ~1% of total miRNA, which looks weird as many miRNAs have robust mRNA targets identified by both in silico and experimental methods. The authors should investigate such a poor matchup rate. Several suggestions below:

a) How were the mRNA and miRNA data matched? Is that each miRNA dataset has its own paired mRNA data? Or mRNA and miRNA are from totally different resources (e.g. different labs, different cells)? The author should have added these pieces of information in the method section to help the reader better understand their study setup.

b) If the authors apply MiTEA on each dataset (DBCG, Oslo2, Micam and Stavanger) and their paired mRNA data independently, how many matchups could be found?

c) What is the p-value and rank of hsa-miR-193b-5p in the MiTEA analysis?

d) The authors should apply other normalization methods on the merged dataset, implement the same sets of analysis in this section (refer to Figure 4 and 5), and compare their performances to that of AQuN.

4. In Figure 6, the authors claimed they observed “an overall increase of statistical confidence”. This seems to be misleading. What does “statistical confidence” refer to here? Does it refer to confidence interval or statistical power? The former one was set to 95% throughout the paper (or maybe 90% as the authors applied one-tailed test) and never changed. If it refers to the latter one, the increase of statistical power could be a directly effect from the increased sample size after merging, regardless of using AQuN or not, making this figure unnecessary. The authors should clarify on this.

Minor Comments:

1. The authors should have always used letters to label each panel of a figure. For example, use Figure 1A, 1B and 1C. But this is inconsistent throughout the paper. In many figures with complicated compositions of panels, they used Top, Middle and Bottom, which were hard for readers to keep on track.

2. When compared to Combat normalization, the authors used Tumor grade, subtype, ER, PR, HER2 and TP53 in addition to batch effects. However, it seems that many of these variables share co-linearity relationships. The authors should investigate this before applying normalization on all of them.

3. The current Result session is a mixture of methods and results. The authors should have separate them better in their corresponding sections.

4. In Figure 5, the authors should clarify what does P98 mean.

5. How was the performance of this normalization approach on the data derived from miRNA sequencing or Nanostring platform? Those are the two major platforms, if not more popular than microarrays, in the filed currently. Or is it suitable on TCGA dataset, which is a much larger cohort? The authors might want to either add a new analysis or comment it in the discussion.

6. In the discussion, “The test evaluated in this analysis is Wilcoxon rank-sum as applied for our differential expression analysis in section 0”. What is “section 0”?

7. The authors should clarify why miR-18a-5p and miR-29b-3p were chosen in the a-priori power analysis.

**Have all data underlying the figures and results presented in the manuscript been provided?**

Reviewer #1: **No: **Table 1 reports data on the top four miRNAs identified by differential expression analysis. Results about all the 33 differentially expressed miRNAs can be report as supplementary material.

Reviewer #2: Yes

PLOS authors have the option to publish the peer review history of their article (what does this mean?). If published, this will include your full peer review and any attached files.

Reviewer #1: No

Reviewer #2: No
---

## [Decision Letter · Decision Letter 1]

6 Dec 2020

Dear Dr. Ben-Elazar,

We are pleased to inform you that your manuscript 'miRNA normalization enables joint analysis of several datasets to increase sensitivity and to reveal novel miRNAs differentially expressed in breast cancer' has been provisionally accepted for publication in PLOS Computational Biology.

Best regards,

Denis Thieffry, PhD

Associate Editor

PLOS Computational Biology

William Noble

Deputy Editor

PLOS Computational Biology

Reviewer's Responses to Questions

**Comments to the Authors:**

Reviewer #1: I find the revised version of the article satisfactory, in particular for having included the evaluation of the impact of the AQuN normalization on batch effect correction through Silhouette index (Fig 10b. and Supplementary Fig1) and having shown the transfection efficiency of hsa-mir-193b. All major points have been sufficiently addressed.

**Have all data underlying the figures and results presented in the manuscript been provided?**

Reviewer #1: Yes

PLOS authors have the option to publish the peer review history of their article (what does this mean?). If published, this will include your full peer review and any attached files.

Reviewer #1: No

---

## [Editor Report · Acceptance letter]

2 Feb 2021

PCOMPBIOL-D-20-00931R1 

miRNA normalization enables joint analysis of several datasets to increase sensitivity and to reveal novel miRNAs differentially expressed in breast cancer

Dear Dr Ben-Elazar,

I am pleased to inform you that your manuscript has been formally accepted for publication in PLOS Computational Biology. Your manuscript is now with our production department and you will be notified of the publication date in due course.

With kind regards,

Alice Ellingham
